# Active, Reactive, and Apparent Power in Dielectrophoresis: Force Corrections from the Capacitive Charging Work on Suspensions Described by Maxwell-Wagner’s Mixing Equation

**DOI:** 10.3390/mi12070738

**Published:** 2021-06-23

**Authors:** Jan Gimsa

**Affiliations:** Department of Biophysics, University of Rostock, Gertrudenstr. 11A, 18057 Rostock, Germany; jan.gimsa@uni-rostock.de; Tel.: +49-381-6020; Fax: +49-381-6022

**Keywords:** DEP force in narrow volumes, capacitor charge cycle, micro-fluidic volumes, DEP trajectories, 2D and 3D modelling, DEP-induced polarizability increase, conditioned polarization, lossy dispersive materials, thermodynamics, Rayleigh’s dissipation function, law of maximum entropy production (LMEP)

## Abstract

A new expression for the dielectrophoresis (DEP) force is derived from the electrical work in a charge-cycle model that allows the field-free transition of a single object between the centers of two adjacent cubic volumes in an inhomogeneous field. The charging work for the capacities of the volumes is calculated in the absence and in the presence of the object using the external permittivity and Maxwell-Wagner’s mixing equation, respectively. The model provides additional terms for the Clausius-Mossotti factor, which vanish for the mathematical boundary transition toward zero volume fraction, but which can be interesting for narrow microfluidic systems. The comparison with the classical solution provides a new perspective on the notorious problem of electrostatic modeling of AC electrokinetic effects in lossy media and gives insight into the relationships between active, reactive, and apparent power in DEP force generation. DEP moves more highly polarizable media to locations with a higher field, making a DEP-related increase in the overall polarizability of suspensions intuitive. Calculations of the passage of single objects through a chain of cubic volumes show increased overall effective polarizability in the system for both positive and negative DEP. Therefore, it is proposed that DEP be considered a conditioned polarization mechanism, even if it is slow with respect to the field oscillation. The DEP-induced changes in permittivity and conductivity describe the increase in the overall energy dissipation in the DEP systems consistent with the law of maximum entropy production. Thermodynamics can help explain DEP accumulation of small objects below the limits of Brownian motion.

## 1. Introduction

Up to the high radio frequency range, polarization processes in complex media, such as particle suspensions, are classified either by their electric mechanisms such as polarizations of electric double layers, of structural interfaces (Maxwell-Wagner) and of molecular dipoles through orientation (Debye) or by the frequency range in which they disperse (alpha, beta, and gamma dispersions) [1,2,3]. At any given frequency, the sum-effect of all polarization processes that can follow the field alternations weakens the field in each half-cycle. With increasing frequency, slower processes disperse and their contributions to medium polarization fade, which is reflected in a frequency-dependent permittivity decrease and the corresponding increase of the complex (specific) conductivity $\underline{\sigma }$ in S/m by up to several orders of magnitude over several frequency decades (complex parameters are underscored). This behavior is described mathematically by the dispersion relation, which considers the field-induced motion of bound charges, which at high frequencies cannot be distinguished from the motion of charges, such as ions, which can already move freely under DC.

In AC fields above approx. 1000 V/m, different AC-electrokinetic movement of freely suspended microscopic objects, such as colloidal particles or biological cells, can be observed when the objects are exposed to homogeneous, inhomogeneous, or rotating fields. While neighboring objects of even polarizability are attracted toward one another, they are attracted or repelled toward electrode surfaces depending on the electrode and object shapes and the object’s polarizability relative to that of the suspension medium. The field-induced forces and torques lead to the translation (dielectrophoresis; DEP) and rotation (electrorotation; ROT) of individual objects or their aggregation in “pearl chains”, which map the electric field, similar to how iron filings map the magnetic field lines.

Since the slow electrokinetic AC effects, which incidentally also occur in the DC field, are uncoupled from the frequency of the inducing field, they are not usually considered polarization mechanisms. However, in DEP or in electro-thermal pumping it is generally accepted that the lower polarizable medium, which may, depending on the frequency of the external field, be either the suspension medium or the object itself, is displaced by higher polarizable medium at sites of higher field strength. It is intuitive that this process increases the overall polarizability of the suspension. The increase in polarizability by field-induced orientation has previously been shown for suspensions of ellipsoidal objects [4].

The description of the AC electrokinetic behavior of objects in the radio-frequency range is usually based on dipole calculations [3,5,6]. In the classical DEP approach, the object is assumed to be small compared to the characteristic distance of the field inhomogeneity. This allows the DEP force calculation to be based on the assumption of induced dipole moments, i.e., the (weakly) inhomogeneous field induces a symmetric polarization. The interaction of the dipole moment with the inhomogeneous field generates unbalanced forces with the two poles of the dipole, leading to DEP (comparable to the dipolophoresis of a dipole in the field of an ion). However, when objects approach other objects or electrodes, a correct description of, for example deformation, orientation, and DEP forces must include induced multipoles. In addition, multi-body problems such as aggregation and pattern formation play a role [7].

Here, the DEP force is derived from the work difference performed when charging two cubes of equal volume containing either a single suspended object or the pure suspension medium. The work difference resulting from exchanging the positions of the two cubes in an inhomogeneous field provides the DEP force. It is shown that DEP increases the overall polarizability and conductivity in a suspension system. Under the condition of constant voltage, which is generally applied to DEP chambers, the increased conductivity is directly related to the increased dissipation of electrical energy, according to the Rayleigh dissipation function. Two homogeneous spheres with complementary properties are used in model calculations, reflecting the two main structural dispersions observed in biological cells. Comparison with the classical electro-quasistatic dipole approximation, which serves as a reference, suggests that the new approach has a slightly higher accuracy for higher volume fractions, which may be important in narrow microchambers [8,9,10,11,12].

## 2. Theory

### 2.1. General Remarks

The overall permittivity or (specific) conductivity of suspensions or emulsions is increased or decreased by introducing objects with higher or lower polarizabilities than the carrier medium. In DEP, media or objects with lower polarizability are replaced by entities with higher polarizability at locations with a high field. The differently polarizable entities can be the suspension media and suspended rigid or emulsified viscous objects in liquid media of different chemical natures or different temperatures (cf. electro-thermal pumps).

The DEP force for an ellipsoidal object is usually derived from the interaction of its induced dipole moment with the inducing field, where the dipole moment is obtained by solving the Laplace equation for electro-quasistatic conditions. Here, the classical DEP force is used as the reference for a different approach, in which the force is derived from the overall electrical work conducted in a charging cycle that permits the field-free transition of the object between two neighboring cuboid boxes in a weakly inhomogeneous field (Figure 1). Without an object, the dielectric properties in the box are given by the external medium. The presence of the object changes the dielectric properties to those of a suspension, which can be calculated by the Maxwell-Wagner mixing equation.

The idea is based on the cycle proposed by Max Born. To circumvent the complicated description of the electrical effects in the transition of charged ions through lipid membranes, Born proposed separating the process into discharging, interface transition, and recharging of the ion.

Version 12 of the Maple software supplication (Maplesoft, Waterloo, ON, Canada) was used to solve and simplify the equations (The main part of the program code is given in the Appendix A The data for the figures was exported to Sigma Plot 11.0 (Systat Software, Inc., San Jose, CA, USA).

### 2.2. Approximation of the Field Gradient

Long, coaxial cylindrical electrodes are a simple means of generating a well-defined inhomogeneous field of radial symmetry. For the symmetry axis in the *z*-direction, the field component in *z*-direction is zero. For a more general description, cuboid medium and suspension boxes are initially assumed. This shape allows an easier transition to future numerical, two-dimensional (2D) models. In 2D models, a length of 1 m is usually assumed for the z-dimension to match the physical units. In the subsequent model calculations, cubic boxes are used without restriction of generality for the cuboid shape.

Along the field gradient, the field strength $\underline{E}={\underline{E}}_{i}$ is increased to ${\underline{E}}_{i+1}$:(1)$${\underline{E}}_{i+1}=\left(1+\gamma \Delta x\right)\underline{E},$$
when moving $\Delta x=x_{i+1}-x_{i}$ in radial direction, from the center of box *i* to the center of box *i +* 1. For simplicity, vectorial parameters are not marked by arrows. With the inverse length parameter $\gamma >0\;m^{-1}$ describing the field inhomogeneity in the *x-y* field plane, the field gradient in radial (*x-*) direction is:(2)$$\frac{\Delta \underline{E}}{\Delta x}=\frac{(1+\gamma \Delta x)\underline{E}-\underline{E}}{\Delta x}=\gamma \underline{E}.$$

Note that the effects of lateral field components on the induced DEP force are neglected in the dipole approximation by assuming objects that are small in comparison to the characteristic distance of the field inhomogeneity, i.e., for objects with typical semiaxis lengths <1/γ. For reasons of symmetry, the induction of a torque by small lateral field components may be excluded at this stage. For γ=0, the external field is homogeneous, the DEP force disappears for symmetry reasons, and compression or elongation forces remain.

### 2.3. DEP Force in the Classical Dipole Approximation

In almost all cases, the electrodes of DEP devices are operated with constant AC voltages that generate the linear field:(3)$$\underline{E}=E_{0}e^{j\omega t},$$
where $j=\sqrt{-1}$, time t and ω being the circular frequency. $E_{0}$ depends on the location in the inhomogeneous field. In x− (radial) direction, the field induces the complex dipole moment:(4)$$\underline{m}=\varepsilon_{0}\varepsilon_{e}V_{0}{\underline{f}}_{CM}\underline{E}=\varepsilon_{0}\varepsilon_{e}V_{0}\left(f_{CM}^{ℜ}+jf_{CM}^{ℑ}\right)\underline{E},$$
where $\varepsilon_{0}$ and $\varepsilon_{e}$ are the permittivity of vacuum and the real part of the relative permittivity of the external medium, which is frequency-independent for aqueous suspension media in the frequency range considered. $V_{0}$ is the volume of the ellipsoidal object with principal semiaxes a, b, and c:(5)$$V_{0}=\frac{4\pi }{3}abc.$$

The complex Clausius-Mossotti factor $\left(f_{CM}\right)$ has both real $\left(f_{CM}^{ℜ}\right)$ and imaginary $\left(f_{CM}^{ℑ}\right)$ parts. For homogeneous objects of the general ellipsoidal shape in the direction of the (oriented) semiaxis a, it is:(6)$${\underline{f}}_{CM}^{a}=f_{CM}^{aℜ}+jf_{CM}^{aℑ}=\frac{{\underline{\varepsilon }}_{i}-{\underline{\varepsilon }}_{e}}{{\underline{\varepsilon }}_{e}+n^{a}\left({\underline{\varepsilon }}_{i}-{\underline{\varepsilon }}_{e}\right)},$$
where $n^{a}$ is the depolarizing coefficient along semiaxis a [6,13]. For spherical objects with $n^{a}=n^{b}=n^{c}=1/3$, Equation (6) reads:(7)$${\underline{f}}_{CM}=f_{CM}^{ℜ}+jf_{CM}^{ℑ}=3\frac{{\underline{\varepsilon }}_{i}-{\underline{\varepsilon }}_{e}}{{\underline{\varepsilon }}_{i}+2{\underline{\varepsilon }}_{e}}.$$

Note that the factors “3” in Equations (5) and (7) are not cancelled out below to avoid confusion for non-spherical objects. The complex relative permittivities of the internal (index *i*) and external (index *e*) media are:(8)$${\underline{\varepsilon }}_{i}=\varepsilon_{i}-j\frac{\sigma_{i}}{\omega \varepsilon_{0}}\;\;\;\;\;\;{\underline{\varepsilon }}_{e}=\varepsilon_{e}-j\frac{\sigma_{e}}{\omega \varepsilon_{0}},$$
where $\sigma_{i}$ and $\sigma_{e}$ are the DC conductivities. Note that according to Maxwell’s equivalent body notion, Equations (6) or (7) can also describe the Clausius-Mossotti factors of confocal shell models for given field frequency and model parameters [3,6,14,15,16]. Solutions for shelled spherical, cylindrical (the 2D representation of a sphere) and ellipsoidal objects are readily available [17,18,19,20,21,22].

Within the dipole approximation, the time-averaged DEP force $F_{x}$ on an ellipsoidal object is expressed by the real part of the scalar product of the induced dipole moment with the gradient of the complex conjugate field ${\underline{E}}^{*}=E_{0}e^{-j\omega t}$. Using Equation (2), we get:(9)$$F_{x}=\frac{1}{2}ℜ\left(\underline{m}\frac{d{\underline{E}}^{*}}{dx}\right)\approx \frac{1}{2}ℜ\left(\underline{m}\frac{\Delta {\underline{E}}^{*}}{\Delta x}\right)=\frac{\gamma }{2}ℜ\left(\underline{m}{\underline{E}}^{*}\right)\;.$$

Moments induced by the weak inhomogeneity of the field are dominated by the dipole moment and neglected. Accordingly, the DEP force is generated by the interaction of the real part of the object’s dipole moment with the inhomogeneous external field. Introducing Equation (4) into Equation (9) leads to the following well-known result:(10)$$F_{x}=\varepsilon_{0}\varepsilon_{e}V_{0}ℜ\left(f_{CM}^{ℜ}+jf_{CM}^{ℑ}\right)\underline{E}\frac{\gamma }{2}{\underline{E}}^{*}=\varepsilon_{0}\varepsilon_{e}V_{0}f_{CM}^{ℜ}\frac{\gamma }{2}E_{0}^{2}.$$

### 2.4. Charging Work for External and Suspension Media Boxes

It is assumed that cuboidal boxes of quadratic cross-section $x^{2}$ in the sheet plane and depths z perpendicular to the sheet plane are bounded by equipotential planes (“virtual electrodes”) of area xz (Figure 1). The boxes of volume:(11)$$V_{box}=x^{2}z,$$
are flooded with a homogeneous field. They can contain either suspension medium or a suspension with a single spherical or ellipsoidal object located in the center of the box. In the absence of the object (marked by ‘e’), box i has the capacitance:(12)$${\underline{C}}_{i}^{e}=\varepsilon_{0}{\underline{\varepsilon }}_{e}\frac{xz}{x}=\varepsilon_{0}{\underline{\varepsilon }}_{e}z,$$
where x in the denominator stands for the box width, i.e., the distance between the virtual electrodes. In the presence of the object the capacitance is:(13)$${\underline{C}}_{i}^{S}=\varepsilon_{0}{\underline{\varepsilon }}_{S}z.$$

The dielectric is a suspension (marked by capital ‘S’) with relative permittivity ${\underline{\varepsilon }}_{S}$. In the calculation of the electric work, effective (RMS) AC voltages or fields are usually employed to eliminate time averaging. For direct comparison with the dipole approximation, the effective values were substituted by field strength peak values:(14)$$E_{0}=\sqrt{\underline{E}{\underline{E}}^{*}}=\sqrt{2}E_{eff}.$$

The electrical work to charge the box with the external medium is:(15)$$W_{e}=\frac{x^{2}E_{eff}^{2}}{2}\;\varepsilon_{0}ℜ\left({\underline{\varepsilon }}_{e}\right)=\frac{x^{2}E_{0}^{2}}{4}\;\varepsilon_{0}\varepsilon_{e}=\frac{V_{box}E_{0}^{2}}{4z}\;\varepsilon_{0}\varepsilon_{e},$$
and for the box with the suspension:(16)$$W_{S}=\frac{V_{box}E_{0}^{2}}{4z}\;\varepsilon_{0}ℜ\left({\underline{\varepsilon }}_{S}\right)=\frac{V_{box}E_{0}^{2}}{4z}\;\varepsilon_{0}\varepsilon_{S}^{ℜ},$$
where ${\underline{\varepsilon }}_{S}$ is obtained from Maxwell-Wagner’s mixing equation for homogeneous ellipsoids [16,23]:(17)$$\frac{3\left({\underline{\varepsilon }}_{S}-{\underline{\varepsilon }}_{e}\right)}{{\underline{\varepsilon }}_{S}+2{\underline{\varepsilon }}_{e}}=\frac{p}{3}\sum_{k=a,b,c}{\underline{f}}_{CM}^{k}.$$

To reduce indexing, homogeneous spherical objects are considered with principal semi-axes of a=b=c. With Equations (7) and (8), we obtain:(18)$${\underline{\varepsilon }}_{S}=\frac{3+2p{\underline{f}}_{CM}}{3-p{\underline{f}}_{CM}}{\underline{\varepsilon }}_{e}=\frac{3+2p{\underline{f}}_{CM}}{3-p{\underline{f}}_{CM}}\varepsilon_{e}-\frac{3+2p{\underline{f}}_{CM}}{3-p{\underline{f}}_{CM}}\left(j\frac{\sigma_{e}}{\omega \varepsilon_{0}}\right).$$

Using Equation (18) for p=0, Equation (16) is transformed into Equation (15). Note that the solution of Equation (17) is straightforward for ellipsoidal objects (Equation (6)). Moreover, such objects are always oriented with their longest axis in the field direction if their polarizability is different from that of the suspension medium [4]. The volume fraction of the single object (Figure 1) is:(19)$$p=\frac{V_{0}}{V_{box}}.$$

Note that Equation (18) considers the volume fraction but not the number of suspended objects.

### 2.5. DEP Force Approximation by a Capacitor-Charging Cycle

Consider the work performed to charge the two boxes *i* and *i +* 1 (Figure 1). Despite the inhomogeneous field, it is assumed that the two boxes *i* and *i +* 1 are flooded by the homogeneous fields ${\underline{E}}_{i}=\underline{E}$ and ${\underline{E}}_{i+1}$, respectively (Equation (1)). The fields are effective in the centers of the boxes at the considered locations of the object. In the first step of the DEP cycle, both boxes are discharged, box *i* in the presence of the object and box *i +* 1 in its absence. The conducted work is (Equations (1), (15) and (16)):(20)$$W_{dis}=-\frac{V_{box}E_{0}^{2}}{4z}\left(ℜ\left({\underline{C}}_{i}^{S}\right)+{\left(1+\gamma \Delta x\right)}^{2}ℜ\left({\underline{C}}_{i+1}^{0}\right)\right)=-\frac{\varepsilon_{0}V_{box}E_{0}^{2}}{4}\left(\varepsilon_{S}^{ℜ}+{\left(1+\gamma \Delta x\right)}^{2}\varepsilon_{e}\right).$$

In the second step, the object is transferred from box *i* to box *i +* 1, without conducting electrical work, before both boxes are recharged in the third step:(21)$$W_{chrg}=\frac{\varepsilon_{0}V_{box}E_{0}^{2}}{4}\left(\varepsilon_{e}+{\left(1+\gamma \Delta x\right)}^{2}\varepsilon_{S}^{ℜ}\right).$$

The overall work in the cycle is:(22)$$\Delta W=W_{chrg}+W_{dis}=\varepsilon_{0}V_{box}\left(\varepsilon_{S}^{ℜ}-\varepsilon_{e}\right)\;\frac{\gamma \Delta x\left(2+\gamma \Delta x\right)}{4}E_{0}^{2}.$$

The summand γΔx in the second parenthesis is neglected, because it is significantly smaller than two for weakly inhomogeneous fields. With Equation (19) we get:(23)$$\Delta W=\varepsilon_{0}\frac{V_{0}}{p}\left(\varepsilon_{S}^{ℜ}-\varepsilon_{e}\right)\;\frac{\gamma \Delta x}{2}E_{0}^{2}.$$

In positive DEP (positive force), an object that is higher polarizable than the suspension medium travels the distance Δx=x between the two box centers in the direction of the field gradient, from the low to the high field. In this case, the sign of the difference in the parenthesis is positive. It is also positive if a low polarizable object starts in the high-field box and travels against the direction of the field gradient in negative DEP. Both positive and negative DEP increase the overall capacitance of the two-box system, which requires increased charging work (ΔW>0). The DEP-force is:(24)$$F_{x}=\frac{\Delta W}{\Delta x}=\varepsilon_{0}\frac{V_{0}}{p}\left(\varepsilon_{S}^{ℜ}-\varepsilon_{e}\right)\;\frac{\gamma }{2}E_{0}^{2}.$$

Note that the object travels in a ‘box-hopping mode’. A more gradual advance could be obtained by assuming a suspension box that is gradually shifted through a long cuboid volume of external medium. This would allow the mathematical transition from the difference quotient to a directional derivative, but would not change the DEP force expression obtained here. After introduction of parameter properties, Equations (23) and (24) can be used to calculate work and force without further simplifications.

However, the relation of Equation (24) to the classical force equation (Equation (10)) is not immediately clear. From considerations of the characteristic properties of suspensions with the mixing equation it is known that the mathematical boundary transition for infinitely small volume fractions (p→0) leads to the solutions for single objects [2,24]. For simplicity, the volume fraction p is eliminated in two steps. In the first step, the summands stemming from ${\underline{\varepsilon }}_{e}$ in Equation (18) are considered:(25)$$\varepsilon_{S}^{ℜ}=ℜ\left(\frac{3+2p{\underline{f}}_{CM}}{3-p{\underline{f}}_{CM}}{\underline{\varepsilon }}_{e}\right)=ℜ\left(\frac{3+2p{\underline{f}}_{CM}}{3-p{\underline{f}}_{CM}}\varepsilon_{e}\right)-ℜ\left(\frac{3+2p{\underline{f}}_{CM}}{3-p{\underline{f}}_{CM}}\left(j\frac{\sigma_{e}}{\omega \varepsilon_{0}}\right)\right).$$

For p→0, the parenthesis term of the second, external-conductivity summand becomes purely imaginary and provides only a reactive, but no active contribution, i.e., no effective DEP force (see below). Neglecting the second summand, we get:(26)$$\varepsilon_{S}^{ℜ}=ℜ\left(\frac{3+2p{\underline{f}}_{CM}}{3-p{\underline{f}}_{CM}}\right)\varepsilon_{e}.$$

And after introduction into Equation (24):(27)$$F_{x}=\varepsilon_{0}\varepsilon_{e}\frac{V_{0}}{p}ℜ\left(\frac{3+2p{\underline{f}}_{CM}}{3-p{\underline{f}}_{CM}}-1\right)\frac{\gamma }{2}E_{0}^{2}.$$

By expanding ${\underline{f}}_{CM}$ with Equations (7) and (8), p is eliminated by the boundary transition p→0:(28)$$F_{x}=3\varepsilon_{0}\varepsilon_{e}V_{0}\frac{\sigma_{i}^{2}+\sigma_{e}\sigma_{i}-2\sigma_{e}^{2}+\omega^{2}\varepsilon_{0}^{2}\left(\varepsilon_{i}^{2}+\varepsilon_{e}\varepsilon_{i}-2\varepsilon_{e}^{2}\right)}{\sigma_{i}^{2}+4\sigma_{e}\sigma_{i}+4\sigma_{e}^{2}+\omega^{2}\varepsilon_{0}^{2}\left(\varepsilon_{i}^{2}+4\varepsilon_{e}\varepsilon_{i}+4\varepsilon_{e}^{2}\right)}\frac{\gamma }{2}E_{0}^{2}=\varepsilon_{0}\varepsilon_{e}V_{0}f_{CM}^{ℜ}\frac{\gamma }{2}E_{0}^{2}.$$

Equation (28) is identical to Equation (10), i.e., the real part of the Clausius-Mossotti factor for spherical homogeneous objects is:(29)$$f_{CM}^{ℜ}=3\frac{\sigma_{i}^{2}+\sigma_{e}\sigma_{i}-2\sigma_{e}^{2}+\omega^{2}\varepsilon_{0}^{2}\left(\varepsilon_{i}^{2}+\varepsilon_{e}\varepsilon_{i}-2\varepsilon_{e}^{2}\right)}{\sigma_{i}^{2}+4\sigma_{e}\sigma_{i}+4\sigma_{e}^{2}+\omega^{2}\varepsilon_{0}^{2}\left(\varepsilon_{i}^{2}+4\varepsilon_{e}\varepsilon_{i}+4\varepsilon_{e}^{2}\right)}.$$

The limiting frequency cases of Equation (29) are known as DEP plateaus (Figure 2):(30)$$\underset{\omega \to 0}{f_{CM}^{ℜ}}=3\frac{\sigma_{i}-\sigma_{e}}{\sigma_{i}+2\sigma_{e}}\;\mathrm{and}\;\underset{\omega \to \propto }{f_{CM}^{ℜ}}=3\frac{\varepsilon_{i}-\varepsilon_{e}}{\varepsilon_{i}+2\varepsilon_{e}}.$$

### 2.6. Electrorotation (ROT) Torque

In DEP and ROT, the induced dipole moment (Equation (4)) interacts with a linear inhomogeneous field and with a rotating (circularly polarized) field, respectively [6]. As the frequency-dependent part of the dipole moment, the Clausius-Mossotti factor (Equations (6) and (7)) reflects the DEP force and the ROT torque, which are proportional to the real and imaginary parts of the Clausius-Mossotti factor, respectively.

Even though the ROT torque can also be derived by an appropriate capacitor-charging approach, this is beyond the scope of the present manuscript. Instead, the known relations of the ROT torque to the imaginary part of the Clausius-Mossotti factor are used directly [3,5,6]. In analogy to Equation (10), the ROT torque reads:(31)$$N_{z}=\varepsilon_{0}\varepsilon_{e}V_{0}f_{CM}^{ℑ}E_{0}^{2}k.$$

Torque is induced around axis z, which is oriented in the direction of unit vector k, perpendicular to the plane of rotation of a circular polarized field. Equation (27) is transformed to
(32)$$N_{z}=\varepsilon_{0}\varepsilon_{e}\frac{V_{0}}{p}ℑ\left(\frac{3+2p{\underline{f}}_{CM}}{3-p{\underline{f}}_{CM}}\right)E_{0}^{2}k.$$

Note that the field gradient $\frac{\gamma }{2}$ (unit: m^−1^) has been dropped from Equations (31) and (32), so that the units of force in Equations (10) and (27) are changing to units of torque.

## 3. Modelling Results and Discussion

### 3.1. Model Parameters

Spherical objects with a radius of a=b=c= 10 µm (Equation (5)) suspended in cubic boxes (side lengths x = 40 µm) were used for the 3D calculations (Figure 1). These dimensions correspond to a volume fraction of p = 0.0654 (Equation (19)), safely below the 0.1-limit required by Maxwell-Wagner’s mixing equation [15]. Here, the lower value can compensate for the different shapes of boxes and objects. Aqueous electrolyte properties with conductivity and relative permittivity of $\sigma_{e}$ = 0.1 S/m and $\varepsilon_{e}$ = 80, respectively, were assumed for the external medium. Two complementary parameter settings were chosen for the properties of the homogeneous objects, combining high dielectric contrast at low and high frequencies with strong dispersion:

I$\sigma_{i}$ = 0.01 S/m, with $\varepsilon_{i}$ = 800, andII$\sigma_{i}$ = 1 S/m with $\varepsilon_{i}$ = 8.

These parameters were chosen to reflect two strong dispersions that qualitatively correspond to the membrane polarization dispersion (i) and bulk conductance dispersion (ii) of biological cells [3,24] or are found in homogeneous objects [9,10]. Accordingly, each of the two objects sweeps two of the four quadrants of the complex plane swept by biological cells [1,5]. If not stated otherwise, work, forces and dissipation were calculated for normalized field strengths of $E_{0}=$ 1 V/m and $E_{1}/E_{0}=1$ (Equation (3)). Note that 10%-field increase per box width corresponds to a field inhomogeneity of $\gamma =0.1/\Delta x=0.1/40\;\mu m=2500\;m^{-1}$ (Equation (2)). For 10 consecutive boxes, the model field strength increases from 1 V/m in box 1 to 2.3579 V/m in box 10. Table 1 lists the values for 10 box centers and 9 box interfaces.

### 3.2. Clausius-Mossotti Factor

The real and imaginary parts of the Clausius-Mossotti factor represent the normalized DEP force and ROT torque, which are usually plotted over frequency (Figure 2A) or in the complex plain (Figure 2B).

The spectra calculated from Equations (27) and (32) before the boundary transition p→0 deviate slightly from the classical model. Presumably, they are more precise in a certain volume fraction range, but this needs to be investigated in more detail. For higher volume fractions, the accuracy is limited by the upper limit of p<0.1 of the mixing equation. For lower volume fractions, it increasingly corresponds to the classical model. Figure 3 shows the relationships for DEP force and ROT torque spectra. Reference spectra were calculated from the spectra of Figure 2 with appropriate prefactors. The visible differences from the reference spectra largely disappear for a box size of x=80 μm, corresponding to p=0.0082.

### 3.3. Conductivity and Dissipation

The volume-specific field absorption or Rayleigh’s dissipation in each box is described by ohmic heating in the absence:(33)$$P_{V}=ℜ\left({\underline{\sigma }}_{e}\right)E_{eff}^{2}=\sigma_{e}\frac{E_{0}^{2}}{2},$$
or presence of the object:(34)$$P_{V}=ℜ\left({\underline{\sigma }}_{S}\right)\frac{E_{0}^{2}}{2}=\sigma_{S}^{ℜ}\frac{E_{0}^{2}}{2}.$$

The expression for the complex conductivity of a suspension of spherical objects was first derived by Wagner [16]:(35)$${\underline{\sigma }}_{S}=\frac{3+2p{\underline{f}}_{CM}}{3-p{\underline{f}}_{CM}}{\underline{\sigma }}_{e}=\frac{3+2p{\underline{f}}_{CM}}{3-p{\underline{f}}_{CM}}\sigma_{e}+\frac{3+2p{\underline{f}}_{CM}}{3-p{\underline{f}}_{CM}}j\omega \varepsilon_{0}\varepsilon_{e}.$$

This is the conductivity version of Equation (18). Applying the same arguments as to Equation (25), we obtain:(36)$$\sigma_{S}^{ℜ}=ℜ\left(\frac{3+2p{\underline{f}}_{CM}}{3-p{\underline{f}}_{CM}}\right)\sigma_{e}.$$

The DC case (and low frequency limit) of this equation was originally given by Maxwell [15] in his treatise, chapter 9, “Conduction through heterogeneous media”:(37)$$\underset{\omega \to 0}{{\underline{\sigma }}_{S}}=\underset{\omega \to 0}{\sigma_{S}^{ℜ}}=\sigma_{S}=\frac{\sigma_{i}+2\sigma_{e}+2p\left(\sigma_{i}-\sigma_{e}\right)}{\sigma_{i}+2\sigma_{e}-p\left(\sigma_{i}-\sigma_{e}\right)}\sigma_{e}.$$

Here, it is obtained for (ω→0) or after introducing the DC limit of the Clausius-Mossotti factor (Equation (30)) into Equation (35).

Figure 4 illustrates Equations (26) and (36), i.e., the real parts of the relative permittivity and conductivity in the box with the suspended object. Comparison of the two equations with Equation (27) shows different frequency-independent prefactors and an additional offset. If this is considered, the right ordinate can be rescaled so that the DEP force (Equation (27)) is represented by the same function plots.

Note that the plotted real part of the conductivity can be interpreted as field-normalized volume-specific dissipation (Equation (34)). Obviously, the presence of the object changes the conductivity of the box medium in such a way that the frequency-dependent absorbance of a certain part of the field energy corresponds to the frequency-dependent work conducted in DEP. The perfect correspondence of the curves indicates a physical background.

### 3.4. Dispersion Relation, Active, Reactive, and Apparent Power

A common representation in impedance research shows the steady decrease of the permittivity of a suspension between frequency plateaus in the so-called dispersion frequency ranges [2,22]. Each decrease in permittivity corresponds to a complementary increase in conductivity. Here, these relationships are described by Equations (18) and (35). The equations can transform into one another using:(38)$${\underline{\sigma }}_{S}=j\omega \varepsilon_{0}{\underline{\varepsilon }}_{S}.$$

Figure 5 illustrates the relationships of the components of the equations for the permittivity and conductivity for suspension of the two model spheres. The apparent permittivity (Equation (18)) and apparent conductivity (Equation (35)) are the sums of their active and reactive components, which are the first and second summands of each of the two equations.

The active components are proportional to the DEP force (Figure 4), suggesting that the reactive component performs no DEP work, but is capacitively stored at the interface of the object out-of-phase with the active component (cf. ROT) and dissipated in the suspension medium, similarly to the reactive power in the peripheral wiring of electrical machines.

### 3.5. Dissipation and Charging Work in the Box Chain

Figure 6 shows the effect of DEP translations on the work of charging and energy dissipation in a chain system of 10 cubic boxes subjected to a constant field gradient according to Equation (1). The field-normalized dissipations in each box were calculated using Equation (33) for the external medium conductivity of 0.1 S/m and the field- normalized values squared of Table 1. Without the object, the dissipation in the first box is 0.1 W/m^3^, which corresponds to the external conductivity of 0.1 S/m at a field strength of 1 V/m. Without the object, the dissipation along the box chain increases with the square of the field strength (Table 1; Figure 6A, left ordinate). The charge work shows the same field strength dependence (Equations (15) and (16)). It is shown on the right ordinate as “apparent relative permittivity”. This parameter assigns an apparent permittivity to each box, so that charging each box with 1 V/m requires the same work as charging the box with the actual field strength in the inhomogeneous field (Table 1). Accordingly, the “apparent relative permittivity” in the first box is 80, the actual permittivity of the suspension medium (cf. left ordinate in Figure 4). For each box, the dissipation and apparent permittivity are plotted for three cases, in the absence and presence of the low (0.01 S/m) or high (1 S/m) conductive sphere. In the calculations, the low frequency limit (Equation (30)) of Equation (36) was used as an example. In the presence of the spheres, the dissipation decreases or increases as indicated for box 6. In DEP, the spheres travel through the box chain against (negative DEP) or in (positive DEP) the direction of the field gradient, depending on their conductivity, i.e., their polarizability relative to that of the suspension medium. The dissipation differences induced by the presence of the spheres increase in the direction of the field gradient, resulting in decreasing and increasing force magnitudes along the DEP trajectories for negative and positive DEP, respectively.

Figure 6B illustrates the DEP effects on the entire box chain considered as a DEP system. Without the object, the field-normalized total dissipation in the chain corresponds to the sum of the 10 boxes (full horizontal line). In the presence of the weakly or strongly conducting sphere, one box contributes to the total dissipation according to Equation (34). The mean chain dissipations (dashed horizontal lines) were calculated assuming a probability 0.1 uniform distribution for the presence of the spheres in each of the boxes. The curves show the dissipation for successive positions of the spheres during DEP translation. In both cases, positive and negative DEP, the field-induced translation leads to an increase in the total dissipation and effective relative polarizability of the DEP system. The latter was calculated by normalizing the sum of the apparent relative permittivities for each sphere position (Figure 6A) to the mean of all positions. The effective relative polarizabilities of one (right ordinate) correspond to the mean dissipations in the entire chain for uniformly distributed (starting) positions of the spheres (dashed horizontal lines). The effective relative polarizabilities of the chain can be used to describe how DEP “conditions” the overall polarizability of the chain system.

In an experimental situation, starting from the random distribution (one), it will increase and rise to a maximum reached when the weakly or strongly conducting (polarizable) object reaches the available positions with the highest or lowest field, respectively. Note that this explanation may be insufficient in real DEP systems, since the presence of the object changes the field distribution, especially near the electrodes, for example by mirror charges. It should also be noted that each field-induced step in or against the direction of the field gradient is directly coupled to the proportional increase or decrease of the active, reactive, and apparent (complex) components plotted in Figure 5.

### 3.6. DEP Force in the Box Chain

The original intention of this manuscript was to derive the DEP force from an alternative approach. The new approach actually provided the classical DEP force expression (Equation (28)) after applying the mathematical boundary transition p→0 to the new force expression Equation (27), which still contains the volume fraction of the object under consideration. Figure 7 compares the two equations. The DEP forces were calculated for the two model spheres at the nine interfaces between the boxes using the DC limit (Equation (30)) of Equation (27).

## 4. General Discussion

### 4.1. Higher Precision for the DEP Force?

The derivation of Equation (28) and the exact agreement with Equation (10) shed new light on the physical background of the required simplifications and the interrelationships of suspension impedance and AC-electrokinetic effects. The new derivation connects the DEP force and DEP translation directly to the changes in dielectric properties in the system. The new DEP force expression (Equation (27)) is based on the frequency dependencies of the active components of the real part of the permittivity (Equation (26)) or conductivity (Equation (36)) of the suspension of a single object. After the p→0 boundary transition, the three equations are identical to the classical DEP force (Equations (10) and (28)) when the appropriate prefactors and offsets are considered. Figure 4 shows slight deviations from the classical model due to additional terms, which are eliminated by the boundary transition. This step cancels possible volume-related polarization properties and reduces Equation (27) to the pure dipole effects. It may be worthwhile to consider the lost terms for a more accurate description of the DEP force in narrow environments [9,10,11,12]. In the classical electro-quasistatic derivation, it is assumed that the field gradient is undisturbed by the presence of the small objects considered. However, for larger objects, i.e., higher volume fractions, the field distributions with and without objects must diverge. This calls for improvements to the model, also to describe the interaction between objects of similar size [7] or of objects at electrode surfaces with mirror charges.

For testing purposes, it might be worthwhile to compare the new model with multipole models [7]. In the future, it might also be interesting to describe the suspension properties in the charge cycle with mixing equations specifically designed for higher volume fractions, such as Hanai’s equation [25,26].

### 4.2. DEP as Conditioned Polarization Process

DEP dissipates field energy. Figure 6 shows how DEP increases the active power dissipation and polarization in the system, regardless of whether the effective conductivity or polarizability of the object is higher or lower than that of the suspension medium. A similar increase has already been described for the electro-orientation of ellipsoidal objects in homogeneous fields (cf. Appendix B in [4]). In electro-orientation, it is the field-induced orientation of the longest axis of homogeneous objects that maximizes the power dissipation in the system, regardless of whether the objects or the medium are higher polarizable. In DEP and electro-orientation, the higher polarizability of the suspension results from field-induced rearrangements of the suspended objects. Relative to the oscillations of the causative field, these processes can be extremely slow. DEP and electro-orientation are not typical polarization processes where the polarization follows the oscillation of the polarizing field. It is therefore proposed that the DEP and possibly AC electrokinetic processes, such as electro-orientation, electro-deformation or field-induced aggregation as “conditioned polarization processes” be considered.

For an explanation, the positive DEP branch of Figure 6B should be expanded in a Gedankenexperiment. The behavior of the many objects in a real suspension system needs to be reproduced by a large number of box chains in a gradient field at constant electrode voltage. In the inhomogeneous field, all the chains should be aligned along DEP trajectories. Before field-on, a single object in each chain is at a random location. After field-on, all the objects move with an increasing velocity in the direction of the field gradient until each object reaches its final position at locations with the highest field, for example, the electrode surface. This state corresponds to the highest conditioned polarizability of the system. The modeling of such processes is possible taking the Stokes friction into account, but they are beyond the scope of this manuscript. Experimentally, the time course of the increase in polarizability should be detectable in microfluidic systems [11].

### 4.3. Relations to the Law of Maximum Entropy Production (LMEP)

It has been shown how DEP synchronously increases the total polarizability and dissipation in the suspension system. It is a physical principle in linear systems that forces act along the (energy) field gradient. Despite the quadratic dependence of the DEP force on the field, for the DEP as well, translation along the field gradient is assumed, which is the fastest way to increase the total polarizability and dissipation in the system in agreement with the LMEP. The LMEP states that a system will select the path or assemblage of paths out of the available paths that minimize the potential or maximize the entropy at the fastest rate [27]. In other words, LMEP demands the maximization of entropy production [28,29,30].

At constant temperature *T*, the entropy production dS/dt in a box is proportional to dissipation [31] according to Equations (33) and (34) in the absence or presence of the object, respectively. The equations are the electrical version of Rayleigh’s dissipation function:(39)$$\Phi =P_{V}V_{box}=\frac{dS}{dt}T,$$
which can be expressed in general terms by products of fluxes and their inducing forces [32,33].

In the experimental situation, a dynamic equilibrium with the thermal forces is established after the DEP translation and rearrangement processes of the objects under the influence of the field are completed. In this equilibrium, the electrical energy dissipation should approach a maximum. It is very likely that the DEP systems allow the study of the balance between the stability of the field-induced structures and the entropy production necessary to keep the structures stable [29]. LMEP could provide a thermodynamic explanation for DEP-induced accumulation of viruses [34] and proteins [35], overcoming the dispersive forces associated with Brownian motion and osmotic segregation, which cannot be explained by current DEP theory [36]. Interestingly, the new force expression shows higher values for positive DEP than the classical dipole approach, a tendency that may be enhanced when mixing equations for higher volume fractions are used [25,26].

However, in micro-chamber experiments with a negative DEP, we observed that the expected DEP end positions were not reached because the DEP forces became too weak to overcome sedimentation, surface friction, and subsequent adhesion for increasing distances to the electrodes.

## 5. Conclusions and Outlook

The DEP force was derived from the electrical charge work of a single-object suspension. Its permittivity has been described by the Maxwell-Wagner mixing equation, which contains the volume fraction but not the number of objects that show the ponderomotive character, i.e., the volume character of the DEP force. The derivation presented can be easily extended to 2D systems and to multishell spheroidal, cylindrical, and ellipsoidal objects with known expressions for their Clausius-Mossotti factors. Along the gradient of an inhomogeneous field, the DEP force was calculated from the differences of the charge work of the system for successive positions of the object. Interestingly, both the increase in capacitive charge work and active energy dissipation provides the frequency dependence of the DEP force when appropriate prefactors and offsets are used. The considerations showed that DEP can be considered as a conditioned polarization process that increases the polarizability, i.e., the total effective permittivity of the DEP system at an extremely low frequency.

The higher polarizability is associated with higher currents, while higher energy dissipation is in accordance with the LMEP. This suggests that DEP movement follows trajectories that increase the polarizability and dissipation at the fastest rate. The application of these principles can simplify the numerical prediction of field-induced forces and object trajectories by simplifying the calculation of the position-dependence of the total conductivity in DEP systems. Moreover, the approach is expected to work not only for modeling DEP trajectories, but also for orientation and aggregation in multi-object systems and for field-induced behavior of objects with irregular shape or internal structures. However, AC electrokinetic forces are induced not only on the suspended objects but also on the suspension media. The superposition of the induced fluid currents on the DEP motion can complicate the prediction of object trajectories in real systems.

Finally, during the completion process of the submission, the author learned of the work of Zheng and Palffy-Muhoray [37], who consider the physical concepts describing electrical energy storage in dissipative materials based on atomic and molecular properties. These authors describe the contradictions between the macroscopic and microscopic approaches and state that the difficulty lies in the partitioning of input power into two distinct components—the dissipation rate and the rate of change of stored energy. The author would like to point to the strong analogy of this description with the contributions of the apparent, active and reactive power components to the DEP presented in this manuscript. Note that the reactive components contributing to the power dissipation can be formally negative.

## Figures and Tables

**Figure 1 micromachines-12-00738-f001:**
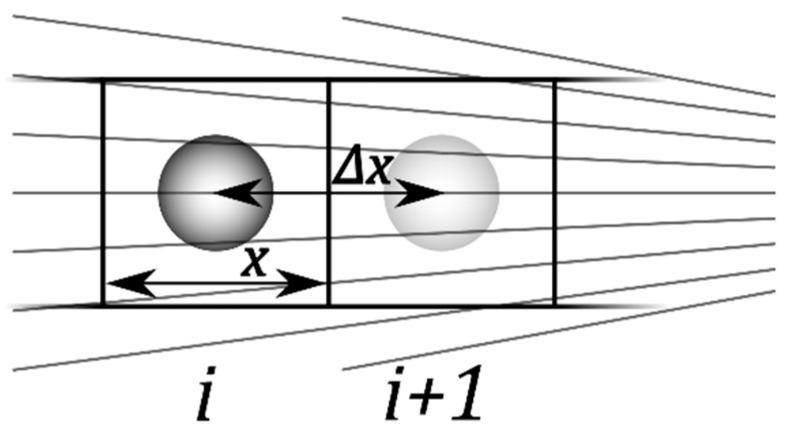
Diagram illustrating the consecutive positions of a spherical object during DEP movement in the inhomogeneous field of a 2D or 3D radial setup. The squares *i* and *i* + 1 represent cuboid boxes with x by x geometry (y=x) in the sheet plane. The field gradient points in the radial direction. Without any limitation in generality for 3D models of microscopic systems, perpendicular to the sheet plane, a depth of z=x or of z= 1 m can be assumed, as is common in 2D models. The distance of travel between the box centers is Δx=x.

**Figure 2 micromachines-12-00738-f002:**
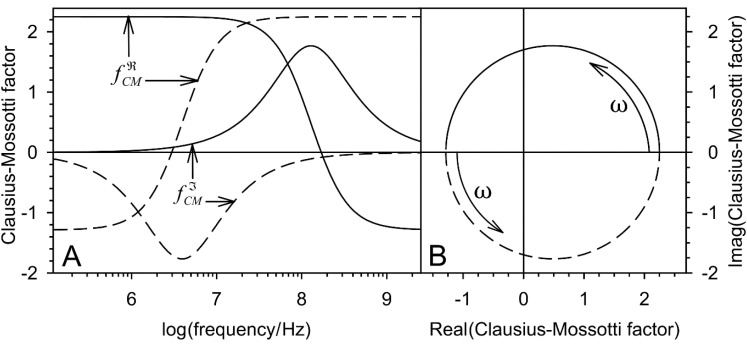
Real and imaginary parts of the Clausius-Mossotti factors of two homogeneous spherical objects in aqueous medium ($\sigma_{e}$ = 0.01 S/m, $\varepsilon_{i}$ = 800) plotted over frequency (**A**) and in the complex plain (**B**) according to the identical Equations (7) or (29). Dashed lines: $\sigma_{i}$ = 0.01 S/m, $\sigma_{i}$ = 800; Full lines: $\sigma_{i}$ = 1 S/m, $\varepsilon_{i}$ = 8). The low and high frequency plateaus (Equations (30)) are clearly visible.

**Figure 3 micromachines-12-00738-f003:**
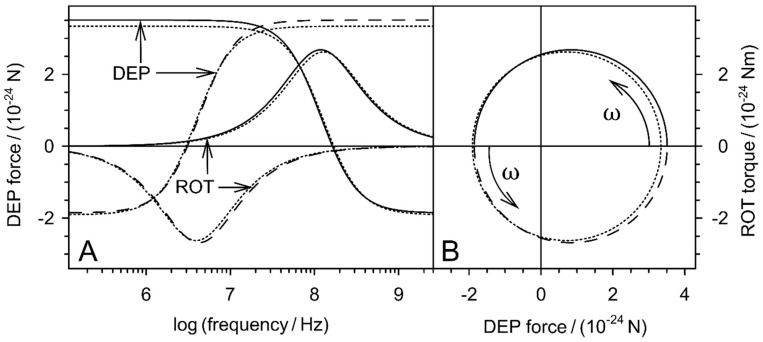
Frequency (**A**) and complex plots (**B**) of DEP-force and ROT-torque spectra of the two spheres of Figure 2 according to Equations (27) and (32) compared with the classical model (dotted lines; Equations (4), (7), (28) and (31)). All spectra were calculated for field strengths of 1 V/m. To obtain corresponding numerical values of forces and moments, the DEP forces were calculated for $\gamma =1\;m^{-1}$.

**Figure 4 micromachines-12-00738-f004:**
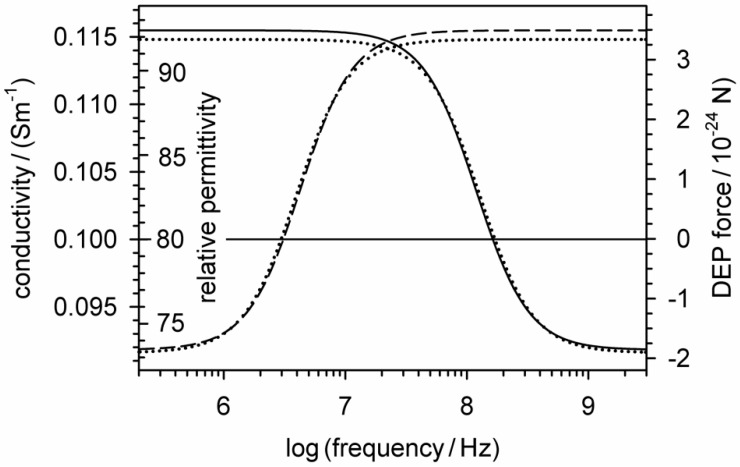
Relationships of the real parts of the relative permittivity and conductivity (Equations (26) and (36)) of the suspension containing two different homogeneous spheres (Dashed lines: $\sigma_{i}$ = 0.01 S/m, $\varepsilon_{i}$ = 800; solid lines: $\sigma_{i}$ = 1 S/m, $\varepsilon_{i}$ = 8) to the induced DEP force (Equation (27)). The DEP forces are identical to those in Figure 3 when using the relative permittivity and conductivity of the suspension medium (80, 0.1 S/m) as a reference for zero DEP force. The dotted lines show the classical DEP force (Equation (10)).

**Figure 5 micromachines-12-00738-f005:**
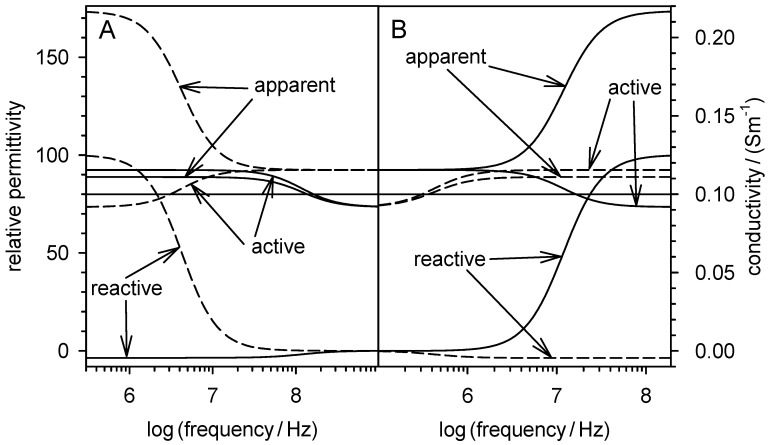
Component plots of the real parts of the relative permittivity ((**A**); Equation (18)) and conductivity ((**B**); Equation (35)) of two suspensions containing a single homogeneous sphere (Dashed lines: $\sigma_{i}$ = 0.01 S/m, $\varepsilon_{i}$ = 800; solid lines: $\sigma_{i}$ = 1 S/m, $\varepsilon_{i}$ = 8). The apparent permittivity and conductivity are the sums of their active and reactive components.

**Figure 6 micromachines-12-00738-f006:**
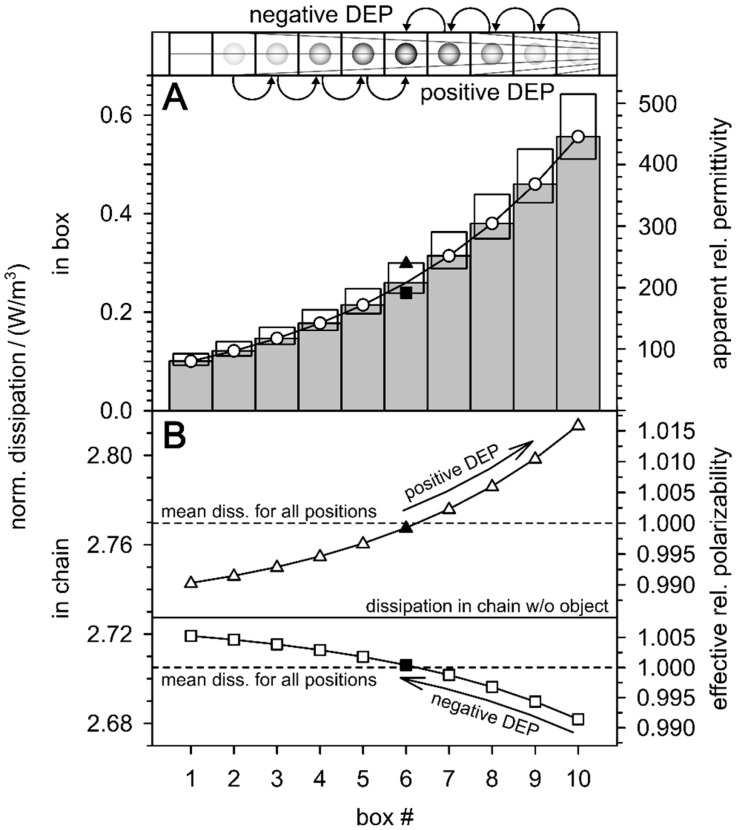
Illustration of DEP-induced changes in field-normalized dissipation in a box chain subjected to an inhomogeneous low-frequency field according to the values of Table 1. (**A**): In the absence of the object, the dissipation in the direction of the field gradient increases with the square of the field strength (circles, gray columns). In the presence of a high (triangles, $\sigma_{i}$ = 1 S/m) or low polarizable sphere (squares, $\sigma_{i}$ = 0.01 S/m), the active dissipation in the box is increased or decreased, respectively. The work of charging has the same field strength dependence (Equations (15) and (16)). It is plotted as “apparent relative permittivity” above the right ordinate (see text). (**B**): Dependence of the sum of dissipation in the box chain system on the positions of the single objects. Arrows denote DEP “trajectories”. The dissipations in the chain system (dashed horizontal lines) correspond to effective relative polarizabilities (right ordinate), which are proportional to the charge work of the whole chain system. Effective relative polarizabilities of one correspond to the average dissipation throughout the chain, for an even distribution of the 10 starting positions for the model spheres (dashed horizontal lines).

**Figure 7 micromachines-12-00738-f007:**
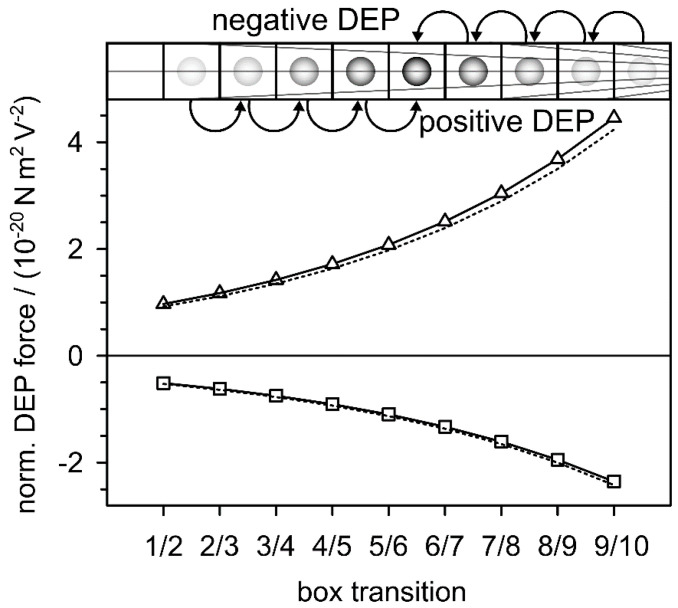
Low frequency plateaus of the DEP forces calculated with Equation (30) for the model spheres with $\sigma_{i}$ = 0.01 S/m (squares) and $\sigma_{i}$ = 1 S/m (triangles) compared with the classical model of Equation (10) (dotted lines). A field inhomogeneity of $\gamma =2500\;m^{-1}$ and normalized field strengths at the box interfaces were assumed (Figure 1, Table 1).

**Table 1 micromachines-12-00738-t001:** Field values used in the calculations.

**Box** i	**Normalized Field** $E_{i}/E_{0}$	**Normalized Field Squared** $E_{i}^{2}/E_{0}^{2}$	**Transition from Box** i→i+1	**Squared Normalized Mean Field at Box Interfaces** ${{}^{\left(\frac{E_{i+1}+E_{i}}{2E_{0}}\right)}}^{2}$
1	1	1		
2	1.1	1.21	1→2	1.1025
3	1.21	1.4641	2→3	1.3340
4	1.331	1.7716	3→4	1.6142
5	1.4641	2.1436	4→5	1.9531
6	1.6105	2.5937	5→6	2.3633
7	1.7716	3.1384	6→7	2.8596
8	1.9487	3.7975	7→8	3.4601
9	2.1436	4.5950	8→9	4.1867
10	2.3579	5.5599	9→10	5.0660

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
