# Peer review of "Active, Reactive, and Apparent Power in Dielectrophoresis: Force Corrections from the Capacitive Charging Work on Suspensions Described by Maxwell-Wagner’s Mixing Equation"

_micromachines, 2021, doi:10.3390/mi12070738_

Round 1
Reviewer 1 Report
The paper is particularly interesting, presenting a new theoretical approach to dielectrophoretic force based on the variation of electrical permittivity with frequency and the Maxwell-Wagner equation. The comparison with the classical solution offers a new perspective on the electrostatic modeling of electrokinetic effects in alternating current and offers a perspective on the relations between active, reactive and apparent power in generating dielectrophoretic force in close connection with the second principle of thermodynamics.
Author Response
Dear reviewer,
I am grateful for your time and efforts as well as for your very positive review.
Sincerely,
Jan Gimsa
Reviewer 2 Report
In this work, authors have presented an alternate expression for DEP force which claims a higher accuracy than classical approach. The work is well presented, however, there are a few points that need to be addressed:
- Authors should make the title brief. It is very long.
- Introduction: Line 86: How does this higher accuracy affect performance and application of the microchambers? what are the critical features governing the higher accuracy? Does this affect design parameters of microchambers?
- Section 2 needs to be more succinct. It is too long for background information. These are known equations and methods and the section can be made more brief.
- Figures 2 and 3 can be combined.
- Have authors evaluated changes in boundary transition (p->0) assumptions?
- It would be great to see a short application of how author's approach improves on a currently existing predictive model of a DEP application (may be a microfluidic application for cell separation?)
- Authors should please proof read the manuscript carefully for spelling and grammatical errors, along with run-on sentences.
Author Response
Dear Reviewer,
Thank you for your time and comments. Below are my responses to your critical points in the order in which you raised them.
- Authors should make the title brief. It is very long.
Reply: You are right, the title “Active, reactive and apparent power in dielectrophoresis: Force corrections from the capacitive charging work on suspensions described by Maxwell-Wagner’s mixing equation” is relatively long. However, it is the result of a longer consideration and I would like to explain each part of the title:
“Active, reactive and apparent power in dielectrophoresis” – this I consider the most important new finding.
“Force corrections” - indicates another finding that could be important for the interpretation of experiments.
“from the capacitive charging work on suspensions” – describes the new way of DEP force derivation applied.
“described by Maxwell-Wagner’s mixing equation” – points to the theoretical basis for the derivation.
- Introduction: Line 86: How does this higher accuracy affect performance and application of the microchambers? what are the critical features governing the higher accuracy? Does this affect design parameters of microchambers?
Reply: The resulting spectra differences are visible in Figs. 3, 4, and 7 for the parameters used in the manuscript. The reasons for the differences are briefly described in the text. They result from the simplifications applied in the classical derivation starting from Laplace's equation. Due to the extremely large parameter space for object and chamber properties, a detailed discussion of the differences and all consequences would clearly go beyond the scope of the present manuscript. It will also require comparison of the new analytical solution with numerical reference solutions (since no other reference exists). A manuscript on such considerations is currently in preparation.
However, together with the revised version of this manuscript, I provide the main part of the Maple program code as an appendix, which allows the comparison of the classical solution with the new one.
- Section 2 needs to be more succinct. It is too long for background information. These are known equations and methods and the section can be made more brief.
Reply: I assume by "section 2" you are referring to "2nd theory"? In principle, you are correct and I could have just used Eq. 10 to compare with the new solution. However, all the parameters introduced before Eq. 10 are needed in the new derivation and had to be defined anyway. Besides, writing down the classical way of derivation gives the reader the opportunity to compare the two ways of derivation and check for the different simplification.
- Figures 2 and 3 can be combined.
Reply: Please note the different ordinate scaling of the figures, which would complicate the composition. As it is, discussion of the figures is easier. There are seven figures in total and no pressure to reduce the number of figures by increasing their complexity.
5. Have authors evaluated changes in boundary transition (p->0) assumptions?
Yes, of course. However, the expansion term for the force equation can be easily derived and is now given in the software code in the appendix.
6. It would be great to see a short application of how author's approach improves on a currently existing predictive model of a DEP application (may be a microfluidic application for cell separation?)
A manuscript on this problem is in preparation.
7. Authors should please proof read the manuscript carefully for spelling and grammatical errors, along with run-on sentences.
The manuscript was checked by a native speaker who is a professional translator and science editor. Therefore, I believe that the language is fine.
Sincerely,
Jan Gimsa